# Representation Finetuning for Continual Learning

## Abstract

The world is inherently dynamic, and continual learning aims to enable models to adapt to ever-evolving data streams. Pre-trained models has shown powerful performance in continual learning. However, since pre-trained models acquire knowledge from static datasets, they still require finetuning to adapt effectively to downstream tasks. Traditional finetuning methods are largely empirical, lack explicit objectives, and still require a relatively large number of parameters. In this work, we introduce **Co**ntinual **Re**presentation Learning(**CoRe**), a novel framework that, for the first time, applies low-rank linear subspace representation finetuning to continual learning. Unlike conventional finetuning approaches, CoRe adopts a learning paradigm with explicit objectives rather than relying on black-box optimization, achieving more efficient parameter utilization and superior performance. Extensive experiments across multiple continual learning benchmarks demonstrate that CoRe not only preserves parameter efficiency but also significantly outperforms existing methods. Our work extends the applicability of representation finetuning and introduces a new, efficient finetuning paradigm for continual learning.

## 1 Introduction

The world is dynamically changing, however, machine learning models are usually trained under the assumption that the training and test data come from the same stationary distribution. When exposed to non-stationary data streams, such models often suffer from Catastrophic Forgetting(Goodfellow et al., 2013)—the tendency to lose previously acquired knowledge when learning new tasks. This challenge has motivated extensive research in continual learning(De Lange et al., 2021)(Masana et al., 2022), whose goal is to enable models to incrementally acquire new knowledge without sacrificing performance on prior tasks.

Recently, continual learning methods built upon pre-trained models have attracted increasing attention(Wang et al., 2022c)(Wang et al., 2022b)(Yu et al., 2024). Pre-trained models, such as ViTDosovitskiy et al. (2020), are extensively trained on large-scale datasets and demonstrate powerful feature extraction capabilities. However, in continual learning scenarios, a domain gap often exists between pretraining datasets and downstream datasets(Zhou et al., 2025). As a result, pre-trained models typically require finetuning to adapt effectively to new tasks. Traditional finetuning methods, such as full-finetuning and parameter-efficient fine-tuning (PEFT)(Houlsby et al., 2019)(Jia et al., 2022), primarily focus on updating model parameters. While effective, these approaches lack interpretability during the learning process, making it difficult to directly explain the role of the updated parameters, and the learning process is largely empirical with unclear objectives. Moreover, they still require a relatively large number of parameters and the parameter efficiency needs to be further improved.

Unlike weight-based finetuning, the recently proposed representation finetuning (ReFT)Wu et al. (2024) directly intervenes on a model's hidden representations, often within a low-rank linear subspace, providing a more flexible and efficient way to adapt models. By enabling task-specific interventions without introducing excessive parameter overhead, ReFT has achieved competitive results across various domains(Liu et al., 2025)(Yin et al., 2024)(Osial et al., 2025)(Huang et al., 2025), particularly in large language models. Nevertheless, despite its effectiveness, ReFT has not yet been explored in the context of continual learning, where controlling representation drift is especially critical.

In this work, we introduce CoRe, the first framework that integrates representation finetuning into continual learning. CoRe applies task-specific interventions within the low-rank subspace of hidden representations, enabling efficient adaptation to new tasks while mitigating catastrophic forgetting. Compared with conventional parameter-efficient fine-tuning methods, CoRe leverages subspace-based interventions to achieve better parameter utilization and superior performance across multiple benchmarks.

Our contributions can be summarized as follows:

- We propose CoRe, the first framework to integrate representation finetuning into continual learning, bridging the gap between representation-level interventions and incremental task adaptation.

- CoRe performs task-specific interventions in the low-rank linear subspace of hidden representations with explicit objectives, ensuring parameter efficiency while improving adaptability.

- Experiments on extensive continual learning benchmarks show that CoRe consistently outperforms existing parameter-efficient fine-tuning methods, achieving state-of-the-art results while maintaining efficiency.

## 2 RELATED WORK

### 2.1 CONTINUAL LEARNING

Traditional continual learning approaches can generally be categorized into three groups: regularization-based, architecture-based, and replay-based methods. Regularization-based approaches(Aljundi et al., 2018)(Serra et al., 2018)(Li & Hoiem, 2017) preserve knowledge from previous tasks by imposing constraints on parameter updates; however, such constraints may also hinder the model's ability to acquire knowledge from new tasks. Architecture-based approaches(Mallya & Lazebnik, 2018)(Serra et al., 2018)(Wang et al., 2020) address this limitation by dynamically modifying the model structure to accommodate new tasks, but this often results in increased memory consumption. Replay-based approaches(Rebuffi et al., 2017)(Buzzega et al., 2020)(Cha et al., 2021), on the other hand, maintain a memory buffer that stores data or knowledge from past tasks, which can be replayed during the training of new tasks to mitigate forgetting. Nevertheless, these methods face challenges such as continually growing memory requirements and potential privacy concerns. Recently, continual learning methods built upon pretrained models have attracted significant attention(Wang et al., 2022c)(Wang et al., 2022b)(Yu et al., 2024). Pretrained models, such as ViTDosovitskiy et al. (2020) and CLIPRadford et al. (2021), are extensively trained on large-scale datasets and exhibit strong feature extraction capabilities. However, a domain gap often exists between the pretraining datasets and the downstream datasets in continual learning scenarios(Zhou et al., 2025). As a result, pretrained models typically require finetuning to adapt effectively to downstream tasks.

### 2.2 REPRESENTATION FINETUNING

Representation finetuning methods intervene in models by modifying the semantic representations of inputs using counterfactual information. ReFTWu et al. (2024) first introduce representation finetuning for adapting large language models (LLMs), achieving competitive results across a wide range of benchmarks. Building on this idea, LoFITYin et al. (2024) identified task-specific critical attention points and trained offsets to modify hidden representations accordingly. IntervMergeOsial et al. (2025) further extend representation finetuning to model merging, employing task-specific interventions to alleviate representational bias. CRFTHuang et al. (2025) apply representation finetuning to Chain-of-Thought reasoning, leveraging information flow analysis to identify and optimize key representations. While representation finetuning has demonstrated competitive performance across various domains, existing approaches have primarily been applied to textual inputs. The potential of presentation finetuning in continual learning has not yet been explored, and further adaptation is required to make it suitable for downstream continual learning tasks.

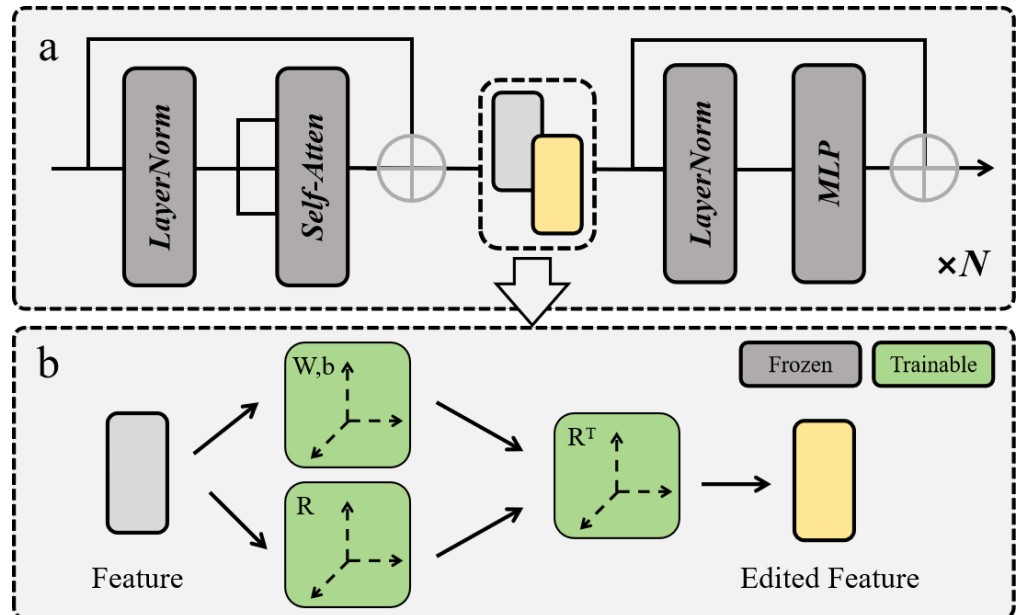

Figure 1: The overall structure of the proposed method. (a) illustrates a standard ViT block, while (b) depicts the implementation of ReFT. Unlike previous finetuning approaches, ReFT directly intervenes in the model by modifying its intermediate features. Specifically, the features are projected into a low-rank subspace via learnable parameters R, w, and b, and then mapped back to the original dimension. Gray areas indicate frozen parameters, while green areas denote trainable components.

## 2.3 PARAMETER-EFFICIENT TUNING

The Parameter Efficient Fine-Tuning (PEFT) method finetunes the pre-trained model by incorporating lightweight modules. During the training process, the pretrained model remains frozen, and model intervention is done only by updating the lightweight modules. Representative methods include AdaptersHoulsby et al. (2019), PromptsJia et al. (2022), and SSFLian et al. (2022). Adapter is a linear module, usually consisting of downsampling, upsampling and activation function. Prompt is a set of learnable vectors that are dynamically added to the hidden layer representation during computation. SSF adds scaling and displacement factors to the model weights, and learns the scaling and displacement factors to adjust the model.

## 3 METHOD

### 3.1 CONTINUAL LEARNING

Given a sequence of tasks $\mathcal{D}_1, \ldots, \mathcal{D}_T$, here The $t$-th task is defined as $\mathcal{D}_t = \{(\boldsymbol{x}_i^t, \boldsymbol{y}_i^t)\}_{i=1}^{m_t}$, where $\mathcal{D}_t$ contains $m_t$ samples $\boldsymbol{x}_i^t$ and their corresponding labels $\boldsymbol{y}_i^t$. Continual learning generally involves three scenarios: task-incremental learning(TIL)(Van de Ven et al., 2022), domain-incremental learning(DIL), and class-incremental learning(CIL). During the continual learning process, for instance in tasks $D_1$ and $D_2$, the input distributions generally differ, i.e.$P(X_1) \neq P(X_2)$, In TIL, both the target distributions $P(Y_1)$ and $P(Y_2)$, as well as the corresponding label spaces $\{Y_1\}$ and $\{Y_2\}$, are distinct. More importantly, the model has access to the task identifier $i$ (task id) during both training and inference. Consequently, at inference time, the model can select the corresponding classifier $\phi_i()$ based on the provided task id. This setting greatly reduces classification conflicts across tasks but depends on the availability of task identifiers, which limits its applicability when task information is unavailable. In DIL, although the data from different tasks originate from distinct domains, they share the same target label distribution, i.e., $P(Y_1) = P(Y_2)$, and a common label space $\{Y_1\} = \{Y_2\}$. The categories remain consistent across domains, but the task identifier is not accessible during inference. This implies that the model cannot rely on task id for prediction, and must instead correctly classify data from unseen domains while overcoming distribution shifts caused by domain variations. CIL

poses the most challenging scenario. In this case, the label distributions $P(Y_1) \neq P(Y_2)$ differ across tasks, and the label space $\{Y_1\} \subseteq \{Y_2\}$ continuously expands as new tasks introduce additional classes. During inference, the task id is not available, and the model must dynamically adapt to the growing set of categories without relying on task information, while retaining the ability to recognize previously learned classes. This greatly increases the risk of catastrophic forgetting.

## 3.2 REPRESENTATION FINETUNING

In large language models (LLMs), taking transformer-based architectures as an example, the model first maps the input text into a semantic representation, which is then propagated through a sequence of block layers progressively learning the semantic representation. Based on the Domain Intervention Interpretation (DII) framework(Geiger et al., 2024), given an initial hidden representation $\boldsymbol{h}_b$ of input text $b$ and representations $\boldsymbol{h}_s$ of counterfactual $s$, the distributed interchange intervention on $\boldsymbol{h}_b$ with the counterfactual source representation $\boldsymbol{h}_s$ could be defined as:

$$\text{DII}\left(\boldsymbol{h}_b, \boldsymbol{h}_s, R\right) = \boldsymbol{h}_b + R^\top \left(R\boldsymbol{h}_s - R\boldsymbol{h}_b\right) \tag{1}$$

Where $R \in \mathbb{R}^{d \times k}$ is a low-rank projection matrix, $d$ is the semantic representation dimension, and $r$ is the rank of the subspace in which we intervene. Eq. 1 provides a principle for counterfactual intervention in the model. Wu et al. (2024) further proposed replacing the explicit counterfactual features with learnable linear matrix $W$ and $b$, introducing an orthogonal constraint that yields the representation editing formulation in LLMs:

$$\Phi\left(\boldsymbol{h}_b, \boldsymbol{h}_s, R, W, b\right) = \boldsymbol{h}_b + R^\top \left(W\boldsymbol{h}_b + b - R\boldsymbol{h}_b\right) \tag{2}$$

## 3.3 CONTINUAL LEARNING WITH REFT

In the context of continual learning with visual inputs, representations requiring intervention can similarly be regarded as counterfactual information. For instance, if the model incorrectly classifies Samoyed's visual feature $\boldsymbol{e}_s$ as spotted dog $\boldsymbol{e}_b$. By applying Eq.2, the feature representation of a Samoyed $\boldsymbol{e}_s$ can be treated as counterfactual information to guide the model's finetuning. As shown in Fig.1, for transformer-based pretrained models, such as ViT, the learned representations can similarly be modified through an learnable linear transformation: $W\boldsymbol{e} + b$ in place of explicit counterfactual. This leads to the following formulation, which removes the dependency on manually constructed counterfactual information:

$$g_\theta(\boldsymbol{e}_b) = \boldsymbol{e}_b + R^\top \left(W\boldsymbol{e}_b + b - R\boldsymbol{e}_b\right) \tag{3}$$

Here, $R$ defines the intervention subspace, while $W$ and $b$ are used to learn the calibration rule. The objective is to make $W\boldsymbol{e}_b + b$ approximate $\boldsymbol{e}_s$, thereby aligning the transformed representation $g_\theta(\boldsymbol{e}_b)$ more closely with the true feature $\boldsymbol{e}_s$. In this way, the original representation $\boldsymbol{e}_b$ is calibrated within a low-rank subspace, rather than being optimized in a black-box manner. With Eq.3, the finetuning of pre-trained model $f_0$ could be described as:

$$f^*(\boldsymbol{x}) = \mathcal{F}\left(f_0(\boldsymbol{x}), \mathcal{D}_1, \theta\right) \tag{4}$$

Where $f^*(\boldsymbol{x})$ is the finetuned model, $f_0$ is the pre-trained model, like ViT, $\theta$ is the trainable parameters of ReFT.

## 4 EXPERIMENTS

### 4.1 IMPLEMENTATION DETAILS

**Benchmark** We evaluate our approach under three continual learning scenarios: task incremental learning, domain incremental learning, and class incremental learning. In the task incremental learning setting, following Yu et al. (2024), we evaluate model performance on a collection of datasets, including AircraftMaji et al. (2013), Caltech101Fei-Fei et al. (2004), CIFAR100Krizhevsky et al. (2009), DTDCimpoi et al. (2014), EuroSATHelber et al. (2019), Flowers102Nilsback & Zisserman (2008), Food101Bossard et al. (2014), MNISTLeCun et al. (2002), OxfordPetParkhi et al. (2012), StanfordCarsKrause et al. (2013), and SUN397Xiao et al. (2010). These datasets cover diverse characteristics: fine-grained datasets such as Aircraft and StanfordCars; broader-coverage

datasets such as EuroSAT; and large-scale scene recognition datasets such as SUN397. For evaluation, EuroSAT and MNIST are divided into 5 tasks with 2 classes per task, while Oxford Pet is split into 8 tasks. In addition, All other datasets are partitioned into 10 tasks, each containing an equal number of classes. In the domain incremental learning setting, following Wang et al. (2022a), we conduct experiments on CDDBLi et al. (2023), CORe50Lomonaco & Maltoni (2017), DomainNetPeng et al. (2019), and OfficeHomeVenkateswara et al. (2017). Among them, CDDB focuses on user behavior recognition and consists of 10 domains; CORe50 contains 50 object classes captured under 11 different lighting and background conditions; DomainNet is the largest and most diverse benchmark for domain shift classification, consisting of 345 categories across 6 highly heterogeneous domains. In this setting, each domain is regarded as a separate task, and each task includes all categories. In the class incremental learning setting, following Zhou et al. (2025), we evaluate our method on CIFAR100Krizhevsky et al. (2009), CUB200Wah et al. (2011), ImageNet-AHendrycks et al. (2021b), ImageNet-RHendrycks et al. (2021a), ObjectNetBarbu et al. (2019), OmniBenchmarkZerroug et al. (2022), and VTABZhai et al. (2019). These datasets include fine-grained recognition benchmarks such as CUB200, challenging subsets of ImageNet such as ImageNet-A and ImageNet-R, and large-scale benchmarks such as OmniBenchmark. In this setting, CIFAR100 and ImageNet-R are split into tasks with 5 classes per task, OmniBenchmark into tasks with 30 classes each, while the remaining datasets are divided into tasks containing 10 classes each. For clarity, we denote the number of classes in each task using the notation 'Inc'. For example, "Inc10" indicates that every contains 10 classes.

Table 1: Performance comparison of different methods under the Task Incremental Learning scenario. All experiments are based on the ViT-B/16-IN21k. Each task consists of 5 classes for DTD and OxfordPet, 2 classes for EuroSAT and MNIST, 20 classes for StandCars and SUN397, and 10 classes for the remaining datasets. The best results are highlighted in bold.

|  | Method | Aircraft | Clatch101 | CIFAR | DTD | EuroSAT | Flowers | Food | MNIST | OxfordPet | StanfordCars | SUN397 |
|---|---|---|---|---|---|---|---|---|---|---|---|---|
| Avg | ViT | 64.50 | 99.01 | 95.14 | 92.57 | 96.34 | 99.86 | 96.23 | 96.49 | 97.09 | 72.15 | 93.25 |
|  | Finetune | 67.94 | 99.36 | 95.03 | 92.39 | 89.60 | 99.81 | 94.12 | 96.38 | 97.01 | 70.66 | 93.11 |
|  | VPT | 69.33 | 99.29 | 98.01 | 90.04 | 89.84 | 99.79 | 96.61 | 94.60 | **97.56** | **79.92** | 93.44 |
|  | SSF | 66.63 | 99.31 | 97.52 | **93.85** | 94.39 | 99.81 | 95.96 | 97.36 | 96.98 | 75.11 | 93.14 |
|  | Adapter | 64.55 | 99.08 | 97.98 | 92.57 | 95.89 | 99.86 | 96.40 | 99.48 | 96.97 | 72.22 | 93.19 |
|  | Ours | **75.09** | **99.43** | **98.15** | 92.71 | **96.43** | **99.87** | **96.77** | **99.60** | 97.00 | 79.41 | **93.48** |
| Last | ViT | 66.16 | 99.25 | 94.76 | 92.98 | 93.26 | **99.89** | 96.06 | 97.37 | **98.15** | 72.89 | 93.24 |
|  | Finetune | 66.07 | 99.20 | 94.61 | 93.19 | 85.33 | 99.84 | 94.28 | 96.31 | 97.33 | 69.63 | 93.11 |
|  | VPT | 69.37 | 99.03 | 97.45 | 90.96 | 86.31 | 99.82 | 96.34 | 94.93 | 97.98 | **80.74** | 93.26 |
|  | SSF | 71.05 | 99.41 | 96.90 | **94.31** | 90.24 | 99.85 | 95.67 | 97.43 | 97.89 | 75.46 | 93.03 |
|  | Adapter | 66.22 | 99.36 | 97.53 | 92.98 | 92.61 | **99.89** | 96.28 | 99.25 | 98.06 | 73.05 | 93.11 |
|  | Ours | **74.89** | **99.62** | **97.59** | 93.35 | **93.15** | **99.89** | 96.45 | 99.39 | 97.93 | 80.40 | **93.40** |

**Baseline** Model finetuning aims to adapt pre-trained models to downstream tasks by updating only a small fraction of parameters compared with the full model. We compare CoRe against several representative parameter-efficient fine-tuning methods, including AdapterHoulsby et al. (2019), PromptJia et al. (2022), and SSFLian et al. (2022). Specifically, Adapter introduces trainable projection layers, typically consisting of down-sampling, up-sampling, and non-linear activation functions, into pre-trained models. Prompt tuning employs a set of learnable vectors that are concatenated with hidden layer representations to modulate model outputs. SSF finetunes models by learning task-specific scaling and shifting parameters applied to model weights. In addition to these PEFT baselines, we also include comparisons with the frozen pretrained model outputs as well as full-parameter finetuning.

**Training Details** We evaluate our method under different continual learning settings, including task incremental learning(TIL), domain incremental learning(DIL), and class incremental learning(CIL). During training, following Zhou et al. (2025), we train the model only on the first task and compute the class-wise feature means, which are then used as the classifier weights. For subsequent tasks, the model parameters remain frozen, and only the prototype-based classifier is updated. The classifier design differs across the three continual learning scenarios. In TIL, we construct a task-specific

classifier for each task. During inference, the task identifier is provided to select the corresponding classifier for prediction. In DIL, following Wang et al. (2022a), we create a separate classifier and a set of k-means cluster centers for each domain during training. At inference time, the input is assigned to the nearest domain center, and the corresponding classifier is used. In CIL, the classifier is dynamically expanded as new classes arrive, and the model must predict over all previously seen classes without any external task information. All experiments are conducted using the ViT-B/16 pretrained on ImageNet-21K, which pretrained on ImageNet21k. For finetuning, we adopt stochastic gradient descent (SGD) as the optimizer with an initial learning rate of 0.05, scheduled by cosine decay. A weight decay of 0.0005 is applied. The batch size is set to 48, and epochs is 20.

**Evaluation Metric** In model performance evaluation, we follow Zhou et al. (2025), using Average accuracy (**Avg**) and Last accuracy (**Last**) to measure the performance of model. Specifically, **Last** denotes the Top-1 accuracy of every task, and **Avg** is the average value of **Last** of all tasks. Mathematically, for the $t$-th task, Average accuracy is calculated as follows: $\mathbf{Avg}_t = \frac{1}{t} \sum_{i=1}^{t} \mathbf{Last}_i$.

Table 2: Performance comparison of different methods under the Domain Incremental Learning scenario. All experiments are conducted using ViT-B/16-IN21k. For all datasets, each task contains the same number of classes. The best results are highlighted in bold.

| Method | OfficeHome Inc65 | | Core50 Inc50 | | CDDB Inc2 | | DomainNet Inc345 | |
|---|---|---|---|---|---|---|---|---|
| | Avg | Last | Avg | Last | Avg | Last | Avg | Last |
| ViT | 74.22 | 64.99 | 77.55 | **61.49** | 76.30 | 72.41 | 50.73 | 40.37 |
| Finetune | 69.06 | 58.15 | 75.86 | 58.22 | 71.98 | 66.64 | 55.31 | 42.54 |
| VPT | 75.30 | 65.95 | 76.20 | 57.20 | 76.51 | 72.42 | 55.32 | 44.09 |
| SSF | 75.48 | 65.80 | 76.42 | 59.02 | 75.39 | 69.34 | 53.57 | 42.50 |
| Adapter | 74.67 | 65.83 | 76.50 | 58.53 | 76.64 | 72.07 | 55.51 | 44.23 |
| Ours | **75.96** | **66.30** | **77.95** | 59.91 | **77.35** | **72.50** | **56.73** | **45.23** |

## 4.2 MAIN RESULTS

We first compare CoRe with various finetuning methods under the task-incremental learning setting. Results are reported in Tab.1. As shown in Tab.1, CoRe achieves competitive performance across diverse datasets. These results demonstrate that CoRe substantially enhances the feature extraction capability of pretrained models across diverse scenarios. On fine-grained benchmarks such as Aircraft and Oxford Pet, CoRe is able to capture subtle intra-class distinctions, while on domain-shift datasets like EuroSAT it adapts to variations in data distribution without sacrificing stability. Moreover, its strong performance on large-scale datasets such as SUN397 highlights CoRe's scalability and robustness when handling complex visual concepts, underscoring its effectiveness as a general solution for task-incremental learning.

We further evaluate different methods under the domain incremental learning (DIL) setting, where each domain contains the same set of categories, and the model must capture the feature variations of identical classes across domain shifts. Following Wang et al. (2022a), we conduct experiments on CDDB, CORe50, DomainNet, and OfficeHome, with results reported in Tab.2. As observed, CoRe consistently achieves superior performance, effectively learning domain-invariant representations while retaining flexibility for domain-specific features. This balance enables better generalization across shifts in appearance, style, or context, demonstrating robustness in dynamic environments with evolving data distributions.

Finally, we assess all methods in the class incremental learning setting, which is widely regarded as the most challenging scenario in continual learning. In this setting, the task identity is unknown during inference, and the model must classify samples among all previously seen classes. This makes the problem substantially harder but also more reflective of real-world applications. Following Zhou et al. (2025), we evaluate the methods on multiple benchmarks, with results summarized in Tab.3. We observe that CoRe outperforms other finetuning methods in most cases. These results confirm that CoRe effectively retains knowledge from previously learned tasks while acquiring new representations in highly challenging settings. By operating in the representation subspace

rather than directly modifying model weights, CoRe mitigates catastrophic forgetting and preserves discriminative features of past classes. This ability to balance stability and plasticity makes CoRe particularly well-suited for real-world continual learning scenarios where task identity is not available at inference.

Table 3: Performance comparison of various finetuning methods under Class Incremental Learning scenario. All experiments are based on ViT-B/16-IN21k. Here, 'IN-R' denotes ImageNet-R datasets, 'IN-A' refers to ImageNet-A datasets, 'ObjNet' represents the ObjectNet dataset, and 'Omni' stands for the OmniBenchmark dataset. For all datasets, each task contains an equal number of classes. The best results are highlighted in bold.

| Method | CIFAR Inc5 | | CUB Inc10 | | IN-R Inc5 | | IN-A Inc10 | | ObjNet Inc10 | | Omni Inc30 | | VTAB Inc10 | |
| | Avg | Last | Avg | Last | Avg | Last | Avg | Last | Avg | Last | Avg | Last | Avg | Last |
|---|---|---|---|---|---|---|---|---|---|---|---|---|---|---|
| ViT | 87.57 | 81.26 | 92.20 | 86.73 | 62.58 | 54.55 | 60.5 | 49.44 | 65.45 | 53.59 | 79.34 | 73.15 | 85.99 | 84.38 |
| Finetune | 87.67 | 81.27 | 91.82 | 86.39 | 70.51 | 62.42 | 61.57 | 50.76 | 61.41 | 48.34 | 73.02 | 65.03 | 87.47 | 80.44 |
| VPT | 88.46 | 82.17 | 91.02 | 84.99 | 68.79 | 60.48 | 60.59 | 48.72 | 67.83 | 54.65 | 81.05 | 74.47 | 86.59 | 83.06 |
| SSF | 87.78 | 81.98 | 91.72 | 86.13 | 68.94 | 60.60 | **62.81** | **51.48** | 69.15 | 56.64 | 80.53 | 74.00 | 85.66 | 81.92 |
| Adapter | 90.65 | 85.15 | 92.21 | 86.73 | 72.35 | 64.33 | 60.53 | 49.57 | 67.18 | 55.24 | 80.75 | 74.37 | 85.95 | 84.35 |
| CoRe(Ours) | **91.32** | **86.04** | **92.51** | **86.90** | **72.52** | **64.40** | 61.64 | 49.05 | **71.00** | **58.59** | **81.50** | **75.04** | **89.42** | **85.65** |

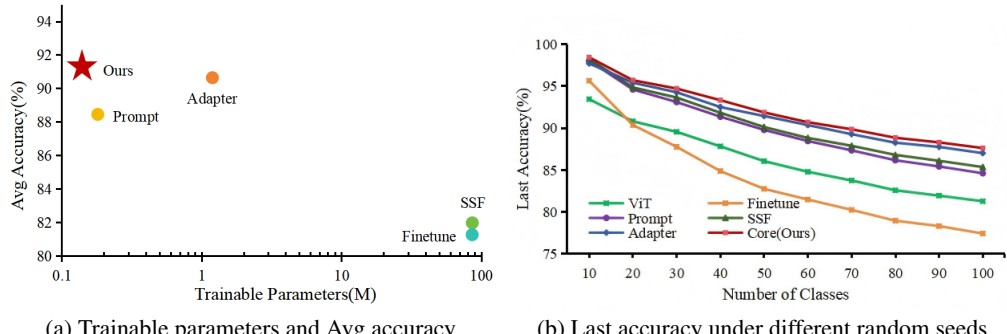

(a) Trainable parameters and Avg accuracy

(b) Last accuracy under different random seeds

Figure 2: Figure (a) present the comparison of trainable parameters and Avg accuracy of finetuning methods, while figure(b) show the average value of Last accuracy of methods under different random seeds. The experiments are conducted on CIFAR Inc10 with ViT-B/16-IN21K.

## 4.3 MODEL ANALYSIS

In the context of model finetuning, parameter efficiency is of critical importance, especially in continual learning scenarios where frequent updates can quickly increase computational and memory costs. We compare CoRe with several representative finetuning methods in terms of the number of trainable parameters and Avg accuracy. The experiments are conducted on CIFAR Inc10 in class increment learning, using ViT-B/16-IN21k as the backbone. The results, illustrated in Fig.2a, show that CoRe achieves the highest average accuracy while using the fewest trainable parameters, efficiently leveraging pretrained representations by updating only task-relevant subspaces. This design enables effective continual adaptation with low computational overhead, highlighting the value of representation-level interventions for scalable and efficient continual learning.

During the training phase of CoRe, we apply a fixed random seed (1993) to shuffle the order of all datasets, ensuring that the training sequence remains unbiased and that results are not influenced by a particular data ordering. To further ensure a fair and robust comparison with other methods, we conduct additional experiments using multiple random seeds 1991, 1992, 1993, 1994, 1995 and report the averaged Last accuracy, as shown in Fig.2b. As observed in Fig.2b, CoRe consistently outperforms competing methods across all seeds, demonstrating not only its superior accuracy but also its robustness to variations in data ordering and initialization. This indicates that CoRe's performance

is stable and reliable, and that its effectiveness does not rely on fortuitous random choices during training.

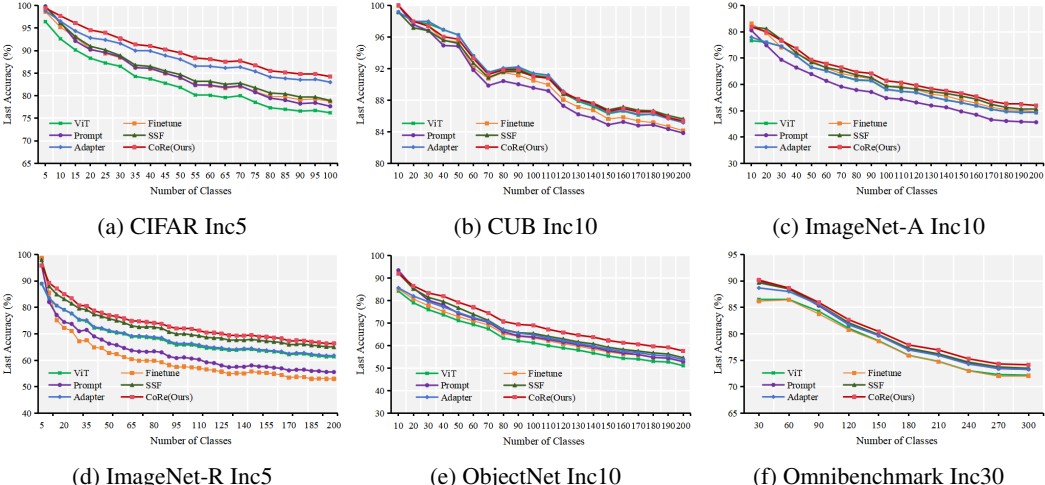

Figure 3: Last accuracy of various finetuning methods under Class Incremental Learning scenario using ViT-B/16-IN1k as the backbone. Each task contains the same number of classes across all datasets. Core consistently outperforms other finetuning approaches even with the ViT-B/16-IN1k architecture.

In real-world scenarios, data is not always uniformly distributed, and continual learning often faces the challenge of class imbalance. We evaluate the performance of different methods under class-imbalanced settings, and experiments are conducted on CIFAR Inc10 with ViT-B/16-IN21k. The results are summarized in Tab.4, where $imb\_factor$ denotes the imbalance ratio. Specifically, the number of samples for class $i$ is computed as $max\_num * (imb\_factor^{(i/(num\_classes))})$ where $max\_num$ is the original number of samples per class in the balanced setting, and $num\_classes$ is the total number of classes. As shown in Tab.4, model performance gradually declines as the imbalance ratio decreases, which highlights that class imbalance poses a substantial challenge for continual learning, leading to biased feature learning and degraded generalization. Nonetheless, CoRe consistently outperforms other methods across varying imbalance ratios, demonstrating its robustness and stability under imbalanced conditions.

In addition, we further evaluate model performance based on ViT-B/16-IN1k. Unlike ViT-B/16-IN21k, this backbone is obtained by first pretrained on ImageNet-21k and then finetuned on ImageNet-1k, which shifts the model's focus. We conduct experiments on CIFAR Inc10 in the class incremental learning setting using ViT-B/16-IN1k as the backbone and compare the Last accuracy across methods. The results presented in Fig.3 indicate that CoRe consistently maintains superior performance across most cases. This suggests that its advantages are not limited to a specific pretrained backbone, but generalize across models with different pretraining histories, confirming its capacity to achieve stable and reliable continual learning under diverse backbone initializations.

## 4.4 ABLATION STUDY

In CoRe, feature representations are updated within a low-rank subspace, and the subspace rank critically affects continual learning performance. We evaluate different ranks on CIFAR Inc10 under the class-incremental learning setting using ViT-B/16-IN21k, with results summarized in Tab.5a. Performance improves as the rank increases from small values, reflecting greater capacity to capture task-specific variations, but degrades when the rank is too high, likely due to redundancy and overfitting. Higher ranks also increase computational cost. Considering this trade-off, we set rank as 8 in this work.

ViT consists of multiple transformer block layers, and finetuning different numbers of layers affects the degree of model intervention. We investigate the impact of inserting ReFT modules into different numbers of blocks on CIFAR Inc10 (Tab.5b). Generally, applying ReFT to more layers improves

Table 4: Performance comparison between CoRe and other finetune methods under different imbalance factors. All experiments are conducted on CIFAR Inc10 with ViT-B/16-IN21k, the best results are highlighted in bold.

| Method | $imb\_factor$ 1 | | $imb\_factor$ 0.5 | | $imb\_factor$ 0.1 | | $imb\_factor$ 0.05 | | $imb\_factor$ 0.01 | |
| | Avg | Last | Avg | Last | Avg | Last | Avg | Last | Avg | Last |
|---|---|---|---|---|---|---|---|---|---|---|
| ViT | 87.13 | 81.26 | 87.07 | 81.14 | 86.70 | 80.88 | 86.51 | 80.47 | 84.70 | 78.07 |
| Finetune | 88.56 | 82.80 | 88.06 | 82.52 | 88.40 | 83.11 | 88.06 | 82.24 | 85.70 | 79.68 |
| VPT | 90.59 | 85.27 | 90.51 | 85.12 | 88.43 | 84.99 | 86.83 | 80.45 | 76.61 | 68.52 |
| SSF | 90.71 | 85.21 | 90.55 | 85.09 | 90.17 | 84.88 | 89.71 | 84.38 | 87.03 | 80.96 |
| Adapter | 92.27 | 87.50 | 91.94 | 87.14 | 89.51 | 84.15 | 88.29 | 82.69 | 85.25 | 79.09 |
| Ours | **92.37** | **87.60** | **92.34** | **87.58** | **91.76** | **86.95** | **91.87** | **86.89** | **89.41** | **83.23** |

performance, with the best results achieved when all 12 blocks are used. Interestingly, inserting ReFT into only one block sometimes outperforms three blocks, indicating a nonlinear relationship between module placement and overall performance. Based on these observations, we insert ReFT into all 12 layers to fully leverage the hierarchical feature representations.

Table 5: Ablation studies on (a)ReFT rank and (b)number of block layer inserted with ReFT. "Reft Rank" in (a) indicates the rank value of every ReFT in model. (b) indicates effect of the number of ViT layers integrated with ReFT. The ViT-B/16-IN21k architecture consists of 12 block layers. Here, "1" indicates that ReFT is inserted into only one block layer, while "12" denotes that ReFT is applied to every block layer. All experiments are conducted on CIFAR Inc10 with ViT-B/16-IN21k.

(a) Effect of ReFT rank.

| Reft Rank | Avg | Last |
|---|---|---|
| 4 | 92.28 | 87.62 |
| 8 | 92.34 | 87.63 |
| 16 | 92.27 | 87.51 |
| 32 | 92.02 | 87.29 |
| 64 | 92.01 | 87.08 |

(b) Effect of the number of finetuned layers

| Layer | Avg | Last |
|---|---|---|
| 1 | 91.95 | 87.25 |
| 3 | 91.59 | 86.61 |
| 6 | 92.16 | 87.40 |
| 9 | 92.17 | 87.59 |
| 12 | 92.24 | 87.63 |

# 5 THE USE OF LLM

In accordance with the ICLR 2026 policy on the use of large language models (LLMs), we disclose that we employed OpenAI GPT-4o for language refinement. The model was prompted with: "I am writing a conference paper, please help me polish this paragraph in an academic style." The LLM was used solely for linguistic refinement and did not generate any new content, experimental results, or analyses. All results were reviewed and verified by the authors, who take full responsibility for the final manuscript.

# 6 CONCLUSION

In this work, we introduced CoRe, the first framework that integrates representation finetuning into continual learning. Unlike conventional parameter-efficient finetuning approaches that intervene primarily at the weight level, CoRe performs task-specific interventions in a low-rank subspace of hidden representations and adopts a learning paradigm with explicit objectives rather than relying on black-box optimization, enabling efficient adaptation to new tasks while mitigating catastrophic forgetting. Through extensive experiments, we demonstrated that CoRe consistently outperforms existing finetuning baselines, achieving state-of-the-art performance while maintaining parameter efficiency. These results highlight the potential of representation-level interventions as an effective and scalable alternative to weight-based adaptation in dynamic learning environments. Our work not only extends the applicability of ReFT to continual learning for the first time but also establishes a new paradigm for efficient fine-tuning in lifelong learning, providing insights for future research on representation-level adaptation in large pre-trained models.

## ETHICS STATEMENT

This work presents a novel algorithm for Continual Learning. Our research is based on publicly available benchmark datasets and does not involve any human subjects, personal data, or other sensitive information. We are not aware of any direct negative ethical impacts or malicious use cases of our work. However, as with any AI technology, we encourage the community to consider the potential for unintended consequences and to use it responsibly.

## REPRODUCIBILITY STATEMENT

To facilitate the reproducibility of our work:

- **Code:** The source code will be made publicly available upon acceptance of this paper.
- **Experiments:** The full experimental setup, including hyperparameter values and training details, is described in Section 4.
- **Data:** All experiments in this paper are conducted on **publicly available benchmark datasets**. A complete list of the datasets used, along with their respective citations, is provided in Section 4

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
