# OpenReview forum: "Representation Finetuning for Continual Learning"
_ICLR.cc/2026/Conference — ICLR 2026 Conference Withdrawn Submission_

### Official Review · Reviewer_c1wY · 2025-10-25

**Soundness:** 2
**Presentation:** 2
**Contribution:** 1
**Rating:** 2
**Confidence:** 4

**Summary:**

The paper proposes a method for continual learning by leveraging low-rank adaptation techniques. The authors adapt existing representation finetuning methods to the continual learning scenario and compare their approach with other state-of-the-art methods. The experimental results demonstrate the effectiveness of the proposed method.

**Strengths:**

The paper addresses the finetuning techniques in continual learning with an emphasis on low rank adaptation. The paper adopted the representation finetuning methods into continual learning scenario and provided comparisons with existing methods to validate its effectiveness.

**Weaknesses:**

1. The method is applying an existing representation finetuning method to the continual learning scenario, the contribution is too incremental.
2. There is no clear motivation for applying the specific finetuning method.

**Questions:**

1. What is the objective of using 'Avg' as the metric?
2. Actually, I suggest the authors to dig deeper into the functionality of the proposed method, e.g. how modules of different tasks can cooperate with each other to boost the performance.

---

### Official Review · Reviewer_A7of · 2025-10-27

**Soundness:** 1
**Presentation:** 1
**Contribution:** 1
**Rating:** 2
**Confidence:** 5

**Summary:**

This manuscript investigated the problem of continual learning with pre-trained models (CL-PTM). The authors introduced Continual Representation Learning (CoRe) that applied low-rank linear subspace representation finetuning for CL-PTM. Specifically, the proposed CoRe adopted the explicit objectives with the tool of counterfactual intervention to achieve more efficient parameter utilization. Experiments with several datasets were conducted to demonstrate the performance of the proposed method.

**Strengths:**

1. This manuscript considered the perspective of the counterfactual intervention, which is interesting and less explored in the context of CL-PTM.

**Weaknesses:**

1. The motivation of the proposed method is not clear. I didn't get why the descriptions in Sections 3.2 & 3.3 were introduced.
2. Some basic concepts of continual learning in Section 3.1 were not accurate, or even wrong.
3. The presentation of this paper was poor. The context of Domain Intervention Interpretation (DII, shown in Section 3.2), which should be the theoretical foundation of this paper, was not well interpreted. The concept was not clear, and the relevant existing studies was not discussed in the Related Work part.
4. The experimental part was incomplete, including the comparisons with baseline methods, ablation studies, etc.

See the Questions part for more details.

**Questions:**

1. I think some hypotheses/concepts regarding the continual learning in Section 3.1 were not accurate, or even wrong. In TIL, we only have assumptions regarding the distinct label spaces Y, without any strong assumptions regarding the label distribution P(Y). In DIL, it seems that we do not need to constrain the same label distribution. As for CIL, we usually do not need to assume different label distributions for P(Y), and the subset assumption of label space Y is also not accurate. I recommend that the authors make a clear description of the basic concepts in continual learning.
2. Sections 3.2 & 3.3 were badly presented. The authors should introduce more details about the proposed method, starting with the motivation for why the authors proposed the descriptions in Eqs (1)-(4).
3. The background of the foundation of Sections 3.2 and 3.3 should be discussed in the Related Work part.
4. The illustration in Figure 1 is less informative. Could the authors provide more details about the methodology?
5. The comparisons with the baseline methods in this manuscript were insufficient. Few representative studies within CL-PTM were discussed. Please refer to the following representative methods as baselines, including prompt-based methods (L2P, DualPrompt), adapter-based (O-LoRA, InfLoRA, BiLoRA). Please refer to [1].
6. Some claims made by the authors were not verified. For example, the authors claimed that the proposed method achieved parameter efficiency. However, I didn't see any empirical results regarding this claim in the experimental part.


References;
[1] Continual Learning with Pre-Trained Models: A Survey. IJCAI 2024

---

### Official Review · Reviewer_bjgc · 2025-10-29

**Soundness:** 2
**Presentation:** 2
**Contribution:** 2
**Rating:** 2
**Confidence:** 3

**Summary:**

This manuscript introduces representation finetuning into continual learning methods based on pre-trained models, enabling task-specific interventions within the low-rank subspace of hidden representations. The authors conduct experimental validation across several datasets and evaluate performance under three continual learning scenarios.

**Strengths:**

(1)	The manuscript introduces representation finetuning methods into continual learning framework. The content is clearly presented and well structured, enhancing the clarity and readability of this manuscript.
(2)	Experiments are conducted across several public datasets, analyzing the effects of subspace rank and data imbalance under different scenarios to evaluate the effectiveness of the proposed method.

**Weaknesses:**

(1)	The analysis of existing pre-trained continual learning methods is insufficient. The authors introduce three typical continual learning methods in the related work. It is suggested to include a comparative analysis of the advantages and limitations of existing methods based on pre-trained models, providing a clearer perspective on pre-training-based continual learning methods.
(2)	The experimental comparisons do not include the latest methods. In Section 4, the authors compare three parameter-efficient finetuning techniques—Adapter, Prompt, and SSF—all developed before 2023. As a proposed continual learning algorithm, it is suggested to include comparisons with recent state-of-the-art methods based on pre-trained models to validate the effectiveness of the proposed approach.
(3)	The experimental evaluation metrics are inadequate. In Section 4.1, the authors use Average accuracy and Last accuracy as evaluation metrics for the proposed algorithm. Given that catastrophic forgetting is a key challenge in continual learning, it would be beneficial to incorporate the model's forgetting rate as an evaluation to enhance the comprehensiveness of the experimental results.
(4)	The authors introduce representation finetuning techniques into continual learning. However, it remains unclear how this approach differs from prompt-based continual learning methods and what specific technical challenges arise in its implementation. It is suggested that the authors provide further clarification in the introduction section.
(5)	The authors formulate a continual learning approach based on representation finetuning in Section 3.3, and how this approach was implemented experimentally is unclear. It is suggested to include a more detailed descriptions of the algorithm’s implementation in the experimental section.
(6)	Some sentences contain redundant expressions—for example, the description of the pre-training dataset in line 275. It is suggested to review the manuscript's language carefully to enhance the overall coherence.

**Questions:**

Please refer to the Weaknesses.

---

### Official Review · Reviewer_o5rm · 2025-11-01

**Soundness:** 1
**Presentation:** 1
**Contribution:** 1
**Rating:** 2
**Confidence:** 3

**Summary:**

This paper introduces CoRe (Continual Representation Learning), a novel framework that applies low-rank linear subspace representation finetuning (ReFT) to continual learning (CL). The authors argue that traditional finetuning methods are empirical, lack explicit objectives, and are parameter-inefficient. CoRe intervenes directly on hidden representations within a low-rank subspace, enabling task-specific adaptation while mitigating catastrophic forgetting. Extensive experiments across task-incremental (TIL), domain-incremental (DIL), and class-incremental (CIL) learning benchmarks demonstrate that CoRe achieves state-of-the-art performance with superior parameter efficiency compared to existing PEFT methods like Adapter, VPT, and SSF.

**Strengths:**

+ The paper is well-written, clearly structured, and easy to follow. Figures and tables are informative and support the narrative effectively. The method is explained with sufficient detail, and the experimental setup is thoroughly described to ensure reproducibility.

**Weaknesses:**

+ No state-of-the-art CL methods are compared, such as L2P, Dulaprompt, Codaprompt, Inflora, sdlora, hideprompt, etc.
+ CIFAR is already included in the ViT pre-trained data, it is better to use dataset like imagenet-r for experiments sec. 4.3-sec. 4.4.

**Questions:**

See the weakness

---

### Note · Authors · 2025-11-30

**Comment:**

Due to some issues in the paper that need to be corrected, we have decided to withdraw manuscript. We sincerely appreciate the reviewers' valuable feedback.

**Withdrawal Confirmation:**

I have read and agree with the venue's withdrawal policy on behalf of myself and my co-authors.